# On the Linguistic Capacity of Real-time Counter Automata

## Abstract

While counter machines have received little attention in theoretical computer science since the 1960s, they have recently achieved a newfound relevance to the field of natural language processing (NLP). Recent work has suggested that some strong-performing recurrent neural networks utilize their memory as counters. Thus, one potential way to understand the sucess of these networks is to revisit the theory of counter computation. Therefore, we choose to study the abilities of real-time counter machines as formal grammars. We first show that several variants of the counter machine converge to express the same class of formal languages. We also prove that counter languages are closed under complement, union, intersection, and many other common set operations. Next, we show that counter machines cannot evaluate boolean expressions, even though they can weakly validate their syntax. This has implications for the interpretability and evaluation of neural network systems: successfully matching syntactic patterns does not guarantee that a counter-like model accurately represents underlying semantic structures. Finally, we consider the question of whether counter languages are semilinear. This work makes general contributions to the theory of formal languages that are of particular interest for the interpretability of recurrent neural networks.

## 1 Introduction

It is often taken for granted that modeling natural language syntax well requires a hierarchically structured grammar formalism. Early work in linguistics established that finite-state models are insufficient for describing the dependencies in natural language data (Chomsky, 1956). Instead, a formalism capable of expressing the relations in terms of hierarchical constituents ought to be necessary.

Recent advances in deep learning and NLP, however, challenge this long-held belief. Neural network formalisms like the long short-term memory network (LSTM) (Hochreiter & Schmidhuber, 1997) have been shown to perform well on tasks requiring structure sensitivity (Linzen et al., 2016), even though it is not obvious that such models have the capacity to represent hierarchical structure. This mismatch raises interesting questions for both linguists and practitioners of NLP. It is unclear what about the LSTM's structure lends itself towards good linguistic representations, and under what conditions these representations might fall short of grasping the structure and meaning of language.

Recent work has suggested that the expressive capacity of LSTMs resembles that of counter machines (Merrill, 2019; Suzgun et al., 2019; Weiss et al., 2018). Weiss et al. (2018) studied LSTMs with fully saturated weights (i.e. the activation functions evaluate to their asymptotic values instead of intermediate rational values) and showed that such models can express simplified counter languages. Merrill (2019), on the other hand, showed that the general counter languages are an upper bound on the expressive capacity of saturated LSTMs. Thus, there seems to be a strong theoretical connection between LSTMs and the counter automata. Merrill (2019); Suzgun et al. (2019); Weiss et al. (2018) also all report experimental results suggesting that some class of counter languages matches the learnable capacity of LSTMs trained by gradient descent.

Taking the counter machine as a simplified formal model of the LSTM, we study the formal properties of counter machines as grammars. We do this with the hope of understanding to what degree counter machines, and LSTMs by extension, have computational properties well-suited for representing the structure of natural language. The contributions of this paper are as follows:

- We prove that general counter machines, incremental counter machines, and stateless counter machines have equivalent expressive capacity, whereas simplified counter machines (Weiss et al., 2018) are strictly weaker than the general class.

- We demonstrate that counter languages are closed under complement, union, intersection, and many other common operations.

- We show that counter machines are incapable of representing the deep syntactic structure or semantics of boolean expressions, even though they can validate whether a boolean expression is well-formed.

- We prove that a certain subclass of the counter languages are semilinear, and conjecture that this result holds for all counter languages.

## 2 DEFINITIONS

### 2.1 GENERAL COUNTER MACHINE

Informally, we can think of counter automata as finite-state automata that have been augmented by a finite number of integer-valued counters. While processing a string, the machine can update the values of the counters, and the counters can in turn inform the machine's state transitions.

Early results in theoretical computer science established that a 2-counter machine with unbounded computation time is Turing-complete (Fischer, 1966). However, restricting computation to be real-time (i.e. one iteration of computation per input) severely limits the counter machine's computational capacity (Fischer et al., 1968). A similar fact holds for recurrent neural networks like LSTMs (Weiss et al., 2018). We study the capabilities of several types of real-time counter automata.

The first counter automaton we introduce is the *general counter machine*. This machine can manipulate the counters by adding or subtracting from them. The other variants that we will go on to define are special cases of this general machine.

For $m \in \mathbb{Z}$, we write $+m$ to denote the function $\lambda x.x + m$. By $\times 0$, we denote the constant zero function $\lambda x.0$.

**Definition 2.1** (General counter machine). A $k$-counter machine is a tuple $\langle \Sigma, Q, q_0, u, \delta, F \rangle$ with

1. A finite alphabet $\Sigma$

2. A finite set of states $Q$

3. An initial state $q_0$

4. A counter update function

$$u : \Sigma \times Q \times \{0,1\}^k \to \left(\{+m : m \in \mathbb{Z}\} \cup \{\times 0\}\right)^k$$

5. A state transition function

$$\delta : \Sigma \times Q \times \{0,1\}^k \to Q$$

6. An acceptance mask

$$F \subseteq Q \times \{0,1\}^k$$

Given an input string $x$ and a counter machine, we perform computation by processing $x$ one token at a time. For each token, we use $u$ to update the counters and $\delta$ to update the state according to the current input token, the current state, and a finite mask of the current counter values. We formalize this in Definition 2.2.

As a preliminary remark on notation, we use $z(x)$ to denote the zero check function

$$z(x) = \begin{cases} 0 & \text{if } x = 0 \\ 1 & \text{otherwise.} \end{cases} \tag{1}$$

Given a vector $\boldsymbol{x}$, we use $z(\boldsymbol{x})$ to represent this function broadcasted over each element of the vector.

**Definition 2.2** (Counter machine computation). Let $\langle q, \boldsymbol{c} \rangle \in Q \times \mathbb{Z}^k$ be a configuration of machine $M$. Upon reading input $x_t \in \Sigma$, we define the transition

$$\langle q, \boldsymbol{c} \rangle \rightarrow_{x_t} \langle \delta(x_t, q, z(\boldsymbol{c})), u(x_t, q, z(\boldsymbol{c}))(\boldsymbol{c}) \rangle.$$

**Definition 2.3** (Real-time acceptance). For any string $x \in \Sigma^*$ with length $n$, a counter machine accepts $x$ if there exist states $q_1, .., q_n$ and counter configurations $\boldsymbol{c}_1, .., \boldsymbol{c}_n$ such that

$$\langle q_0, \boldsymbol{0} \rangle \rightarrow_{x_1} \langle q_1, \boldsymbol{c}_1 \rangle \rightarrow_{x_2} .. \rightarrow_{x_n} \langle q_n, \boldsymbol{c}_n \rangle \in F.$$

**Definition 2.4** (Real-time language acceptance). A counter machines accepts a language $L$ if, for each $x \in \Sigma^*$, it accepts $x$ if and only if $x \in L$.

We denote the set of languages that are acceptable in real time by a general counter machine as CL. We will use the terms "accept" and "decide" interchangeably, as accepting and deciding a language are equivalent for real-time automata.

## 2.2 COUNTER MACHINE VARIANTS

Now, we can can consider various restrictions of the general counter machine, and the corresponding classes of languages acceptable by such automata.

First, we present the *simplified counter machine* discussed by Weiss et al. (2018). The counter update function in the simplified counter machine has two important constraints compared to the general machine. First, it can only be conditioned by the input symbol at each time step. Second, it can only increment or decrement its counters instead of being able to add or subtract arbitrary constants.

**Definition 2.5** (Simplified counter machine). A counter machine is simplified if $u$ has the form

$$u : \Sigma \rightarrow \{-1, \ +0, \ +1, \ \times 0\}^k.$$

Another variant that we consider is the *incremental counter machine*. This machine also is constrained to have only increments and decrements on its counters, but the counter update function is allowed to depend on the state and counter value.

**Definition 2.6** (Incremental counter machine). An counter machine is incremental if $u$ has the form

$$u : \Sigma \times Q \times \{0, 1\}^k \rightarrow \{-1, \ +0, \ +1, \ \times 0\}^k.$$

Finally, we define a *stateless* variant of the counter machine. Removing state from the counter machine is equivalent to allowing it to only have one state $q_0$.

**Definition 2.7** (Stateless counter machine). A counter machine is stateless if

$$Q = \{q_0\}.$$

## 3 COUNTER LANGUAGE HIERARCHY

### 3.1 SIMPLIFIED COUNTER LANGUAGES

Our first result relating counter classes is to show that the simplified counter languages are a proper subset of the general counter languages. The weakness of the simplified machine is that the update function is conditioned only by the input symbol. Thus, languages like $a^m b^{2m}$, which require switching counting behavior, cannot be decided correctly. We formalize this in Theorem 3.1.

**Theorem 3.1** (Weakness of SCL). *Let* SCL *be the set of languages acceptable in real time by a simplified counter machine. Then,*

$$\text{SCL} \subset \text{CL}.$$

*Proof.* Consider the language $a^m b^{2m}$. This is trivially acceptable by a 1-counter machine that adds 2 for each $a$ and subtracts 1 for each $b$. On the other hand, we shall show that it cannot be accepted by any simplified machine. Assume by way of contradiction that such a simplified machine $M$ exists.

Tracking the ratio between $a$'s and $b$'s requires infinite state. Thus, the counters of $M$, as opposed to the finite state, must encode whether $2m = l$ for strings of the form $a^m b^l$. Let $c$ be the value of some counter in $M$. We can decompose $c$ into the update contributed by $a$'s and the the update contributed by $b$'s as follows:

$$c = mu_a + lu_b, \tag{2}$$
$$u_a, u_b \in \{-1, 0, 1\}. \tag{3}$$

Exhausting all the possible functions that $c$ can compute, we get

$$c \in \{0, \pm m, \pm l, \pm(m + l), \pm(m - l)\} \tag{4}$$
$$z(c) \in \{0, \mathbb{1}_{m>0}, \mathbb{1}_{l>0}, \mathbb{1}_{m+l>0}, \mathbb{1}_{m-l\neq 0}\}. \tag{5}$$

We ignore the first four options for $z(c)$, as they clearly do not relate $m$ to $l$. The final option checks whether the ratio is 1, not 2. Thus, $z(c)$ cannot distinguish whether $2m = l$.  $\square$

Note that this argument breaks down if we allow the counter update to depend on the state. In that case, we can build a machine that has two counters and two states: $q_0$ adds 1 to the first counter while it reads $a$, and then decrements the first counter and increments the second counter when it reads $b$. When the first counter is empty and the second counter is not empty, $q_0$ transitions to $q_1$, which decrements the second counter. We accept if and only if both counters are 0 after $x_n$.

## 3.2 INCREMENTAL COUNTER LANGUAGES

Unlike the simplified counter machine, the incremental machine has the same linguistic capacity as the general machine. We can simulate each counter on a general machine with a finite amount of overhead. This provides a reduction from general to incremental machines.

**Theorem 3.2** (Generality of ICL). *Let* ICL *be the set of languages acceptable in real time by an incremental counter machine. Then,*
$$\text{ICL} = \text{CL}.$$

*Proof.* Let $d$ be the maximum that is ever added or subtracted from a counter $c$ in $M$. We simulate $c$ in $M'$ using a counter $c'$ and a value $q \in \mathbb{Z} \mod d$ encoded in finite state. We will implement a "ring-counter" encoding of $c$ such that

$$c' = \lfloor c/d \rfloor$$
$$q = c \mod d.$$

To simulate a $\times 0$ update on $c$, we apply $\times 0$ to $c'$, and transition state such that $q := 0$.

To simulate a $+m$ update on $c$ for some $m \in \mathbb{Z}$, we first change state such that $q := (q + m) \mod d$. Next, we apply the following update to $c'$:

$$\begin{cases} +1 & \text{if } q + m \geq d \\ -1 & \text{if } q + m < 0 \\ +0 & \text{otherwise.} \end{cases} \tag{6}$$

We can compute $z(c)$ by checking whether $z(c') = 0$ and $q = 0$.  $\square$

## 3.3 STATELESS COUNTER LANGUAGES

Similarly, restricting a counter machine to be stateless does not weaken its expressive capacity. We show how to reduce an arbitrary stateful machine to a stateless machine that has been augmented with additional counters. The key idea here is that we can use the additional counters as a one-hot vector that tracks the state of the original machine.

**Theorem 3.3** (Generality of $\tilde{\text{Q}}$CL). *Let* $\tilde{\text{Q}}$CL *be the set of languages acceptable in real time by a stateless counter machine. Then,*
$$\tilde{\text{Q}}\text{CL} = \text{CL}.$$

*Proof.* We define a new stateless machine $M'$ to simulate $M$ by adding a $|Q|$-length vector of new counters called $\boldsymbol{q}'$.

Let $\boldsymbol{\omega}(i)$ denote the $|Q|$-length one-hot vector encoding $i$, i.e. $[\boldsymbol{\omega}(i)]_i = 1$, and all other indices are 0. Note that we consider $\boldsymbol{\omega}(0) = \boldsymbol{0}$.

At initialization, $\boldsymbol{q}'$ encodes the initial state since $\boldsymbol{q}' = \boldsymbol{0} = \boldsymbol{\omega}(0)$. Furthermore, we define the invariant that, at any given time, $\boldsymbol{q}' = \boldsymbol{\omega}(i)$ for some state $i$. Thus, the additional counters now encode the current state.

Let $\boldsymbol{x}\|\boldsymbol{y}$ denote the concatenation of vectors $\boldsymbol{x}$ and $\boldsymbol{y}$. We define the acceptance mask in $M'$ as

$$F' = \{\langle q_0, \boldsymbol{b}\|\boldsymbol{\omega}(i)\rangle \mid \langle q_i, \boldsymbol{b}\rangle \in F\}. \tag{7}$$

An analogous transformation allows us to update the counters inherited from $M$. The last step is to properly update the new counters $\boldsymbol{q}'$. For each transition $\delta(x_t, q_i, \boldsymbol{b}) = q_j$ in $M$, we update $\boldsymbol{q}'$ by adding $-\boldsymbol{\omega}(i) + \boldsymbol{\omega}(j)$. This ensures that the updated value of $\boldsymbol{q}'$ is one-hot since

$$\boldsymbol{\omega}(i) + \big(-\boldsymbol{\omega}(i) + \boldsymbol{\omega}(j)\big) = \boldsymbol{\omega}(j). \tag{8}$$

$\square$

### 3.4 SUMMARY

The general counter machine, incremental counter machine, and stateless counter machine all converge to the same linguistic capacity, which we call CL.

The simplified counter machine defined by Weiss et al. (2018), however, has a linguistic capacity SCL that is strictly weaker than CL.

## 4 COUNTER CLOSURE PROPERTIES

Another way to understand the counter languages is through their closure properties. It turns out that the real-time counter languages are closed under a wide array of common operations, including complement, intersection, union, set difference, and symmetric set difference. The general result in Theorem 4.1 implies these closure properties, as well as many others.

**Theorem 4.1** (General set operation closure). *Let $P$ be an $m$-ary operation over languages. If there exists an $m$-ary boolean function $p$ such that*

$$\mathbb{1}_{P(L_1,..,L_m)}(x) = p\big(\mathbb{1}_{L_1}(x),..,\mathbb{1}_{L_m}(x)\big),$$

*then* CL *and* SCL *are both closed under* $P$.

*Proof.* First, we construct counter machines $M_1,..,M_m$ that decide the counter languages $L_1,..,L_m$. We define a new machine $M'$ that, on input $x$, simulates $M_1,..,M_m$ in parallel, and accepts if and only if

$$p(M_1(x),..,M_m(x)) = 1. \tag{9}$$

$\square$

Let $\Lambda$ be a placeholder for either CL or SCL. Let $L_1, L_2 \in \Lambda$. By Theorem 4.1, $\Lambda$ is closed under the following operations:

$$\Sigma^* \setminus L_1 \tag{10}$$
$$L_1 \cap L_2 \tag{11}$$
$$L_1 \cup L_2 \tag{12}$$
$$L_1 \setminus L_2 \tag{13}$$
$$(L_1 \setminus L_2) \cup (L_2 \setminus L_1). \tag{14}$$

## 5 HIERARCHICAL EXPRESSIONS

We now study the ability of counter machines to represent the language $L_m$ (Definition 5.1). Like natural language, $L_m$ has a deep structure recursively composed from hierarchical constituents.

**Definition 5.1** ($L_m$; Fischer et al., 1968). For any $m$, let $L_m$ be the language generated by:

```
<exp> -> <VALUE>
<exp> -> <UNARY> <exp>
<exp> -> <BINARY> <exp> <exp>
..
<exp> -> <m-ARY> <exp> .. <exp>
```

Surprisingly, Fischer et al. (1968) shows that, by implementing Algorithm 1, even a 1-counter machines can decide $L_m$ in real time. Algorithm 1 uses a counter to keep track of the depth at any given index. If the depth counter reaches $-1$ at the end of the string, the machine has verified that the string is well-formed. We define the arity of a `<VALUE>` as 0, and the arity of an `<m-ARY>` operation as $m$.

---
**Algorithm 1** Deciding $L_m$ (Fischer et al., 1968)

---
1: **procedure** DECIDE($x$)
2:     $c \leftarrow 0$
3:     **for each** $x_t \in x$ **do**
4:         $c \leftarrow c + \text{ARITY}(x_t) - 1$
5:     **return** $c = -1$

---

### 5.1 SEMANTIC EVALUATION AS STRUCTURE SENSITIVITY

While Algorithm 1 decides $L_m$, we observe that it is agnostic to the deep structure of the input in that it does not represent the dependencies between tokens. This means that it could not be used to evaluate these expressions, for example. Based on this observation, we prove that no counter machine can evaluate boolean expressions due to the deep structural sensitivity that semantic evaluation (as opposed to syntactic acceptance) requires. We view boolean evaluation as a simpler formal analogy to evaluating the compositional semantics of natural language.

To be more formal, consider an instance of $L_2$ with values $\{0, 1\}$ and binary operations $\{\wedge, \vee\}$. We assign the following semantics to the terminals:

$$[\![0]\!] = 0 \quad [\![1]\!] = 1 \tag{15}$$

$$[\![\wedge]\!] = \lambda pq.\ p \wedge q \tag{16}$$

$$[\![\vee]\!] = \lambda pq.\ p \vee q. \tag{17}$$

Furthermore, our semantics evaluates each nonterminal by applying the denotation of each syntactic argument to the semantic arguments of the operation. For example,

$$[\![\vee 01]\!] = [\![\vee]\!]([\![0]\!], [\![1]\!]) = 0 \vee 1 = 1. \tag{18}$$

We also define semantics for non-constituent prefixes via function composition. For example,

$$[\![\vee\vee]\!] = [\![\vee]\!] \circ [\![\vee]\!] = \lambda pqr.\ p \vee q \vee r. \tag{19}$$

Finally, we define the language $B$ as the set of expressions $x$ where $[\![x]\!] = 1$ under these semantics.

**Theorem 5.1** (Weak evaluation). *For any $k$, a real-time $k$-counter machine cannot decide $B$.*

*Proof.* Assume by way of contradiction that such an evaluation can be performed. We consider an input $x$ that contains a prefix of $p$ operators followed by a suffix of $p + 1$ values. For the machine to evaluate $x$ correctly, the configuration after $x_p$ must encode which boolean function $x_p$ specifies.

However, a counter machine with $k$ counters only has $O(p^k)$ configurations after reading $p$ characters. We show by induction over $p$ that an $p$-length prefix of operators can encode $\geq 2^p$ boolean functions. Since the counter machine does not have enough configurations to encode all the possibilities, we reach a contradiction.

**Base Case**  With $p = 0$, we have a null prefix followed by one value that determines $[\![x]\!]$. There-fore, we can represent exactly 1 ($2^0$) function, which is the identity.

**Inductive Case**  The expression has a prefix of operators $x_{1:p+1}$ followed by values $x_{p+2:2p+3}$. We decompose the semantics of the full expression to

$$[\![x]\!] = [\![x_1]\!]( [\![x_{2:2p+2}]\!], [\![x_{2p+3}]\!]). \tag{20}$$

Since $[\![x_{2:2p+2}]\!]$ has a prefix of $p$ operators, we apply the inductive assumption to show it can represent $\geq 2^p$ boolean functions. Define $f$ as the composition of $[\![x_1]\!]$ with $[\![x_{2:2p+2}]\!]$. There are two possible values for $f$: $f_\wedge$, obtained when $x_1 = \wedge$, and $f_\vee$, obtained when $x_1 = \vee$. We complete the proof by verifying that $f_\wedge$ and $f_\vee$ are necessarily different functions.

To do this, consider the minimal sequence of values that will satisfy them according to a right-to-left ordering of the sequences. For $f_\wedge$, this minimal sequence ends in 1, whereas for $f_\vee$ it must end in a 0. Therefore, $f$ can have 2 unique values for each value of $[\![x_{2:2p+2}]\!]$. Thus, a $p+1$-length sequence of prefixes can encode $\geq 2 \cdot 2^p = 2^{p+1}$ boolean functions.

$\square$

Theorem 5.1 shows how counter machines cannot represent certain hierarchical dependencies, even when the generated language is within the counter machine's weak expressive capacity. This is analogous to how CFGs can weakly generate Dutch center embedding (Pullum & Gazdar, 1980), even though they cannot assign the correct cross-serial dependencies between subjects and verbs (Bresnan et al., 1982).

## 6  SEMILINEARITY

Semilinearity is a condition that has been proposed as a desired property for any formalism of natural language syntax (Joshi et al., 1990). Intuitively, semilinearity ensures that the set of string lengths in a language is not unnaturally sparse. Regular, context-free, and a variety of mildly context-sensitive languages are known to be semilinear (Joshi et al., 1990). The semilinearity of CL is an interesting open question if we aim to understand the abilities of counter machines as grammars.

### 6.1  DEFINITION

We first define semilinearity over sets of vectors before considering languages. To start, we introduce the notion of a linear set:

**Definition 6.1** (Linear set). A set $S \subseteq \mathbb{N}^k$ is linear if there exist $\boldsymbol{W} \in \mathbb{N}^{k \times m}$ and $\boldsymbol{b} \in \mathbb{N}^k$ such that
$$S = \{\boldsymbol{n} \in \mathbb{N}^m \mid \boldsymbol{W}\boldsymbol{n} + \boldsymbol{b} = \boldsymbol{0}\}.$$

Semilinearity, then, is a weaker condition that specifies that a set is made up of a finite number of linear components:

**Definition 6.2** (Semilinear set). A set $S \subseteq \mathbb{N}^k$ is semilinear if it is the finite union of linear sets.

To apply this definition to a language $L$, we translate each sentence $x \in L$ into a vector by taking $\Psi(x)$, the Parikh mapping of $x$. The Parikh mapping of a sentence is, in more familiar machine learning terms, just its bag of tokens representation. For example, the Parikh mapping of $abaa$ with respect to $\Sigma = \{a, b\}$ is $\langle 3, 1 \rangle$.

**Definition 6.3** (Semilinear language). A language $L$ is semilinear if $\{\Psi(x) \mid x \in L\}$ is semilinear.

### 6.2  SEMILINEARITY OF COUNTER LANGUAGES

While we do not prove that the general counter languages are semilinear, we do prove it for a dramatically restricted subclass of the counter languages. We define $\tilde{\mathrm{Q}}$SCL as the set of language acceptable by a counter machine that is *both* simplified (Definition 2.5) *and* stateless (Definition 2.7), and show that this class is indeed semilinear.

**Theorem 6.1** (Semilinearity of $\tilde{\text{Q}}$SCL). *For all $L \in \tilde{\text{Q}}\text{SCL}$, $L$ is semilinear.*

*Proof.* We express $L$ as

$$L = \bigcup_{\boldsymbol{b} \in F} \{x \mid \boldsymbol{c}(x) = \boldsymbol{b}\} = \bigcup_{\boldsymbol{b} \in F} \bigvee_{i=1}^{k} \{x \mid c_i(x) = b_i\}. \tag{21}$$

Since semilinear languages are closed under finite union and intersection, the problem reduces to showing that $\{x \mid c_i(x) = b_i\}$ is semilinear. We apply the following trick:

$$\{x \mid c_i(x) = b_i\} = \Sigma^* \| Z \| L(\boldsymbol{b}, i) \tag{22}$$

where $Z$ is the set of all tokens that set counter $i$ to 0, and $L(\boldsymbol{b}, i)$ is the set of suffixes after the last occurence of some token in $Z$, for ever string in $L$. Since semilinear languages are closed under concatenation, and $\Sigma^*$ and the finite language $Z$ are trivially semilinear, we just need to show that $L(\boldsymbol{b}, i)$ is semilinear. Counter $i$ cannot be set to zero on strings of $L(\boldsymbol{b}, i)$, so we can write

$$b_i = c_i(x) = \sum_{t=1}^{n} u_i(x_t) = \sum_{\sigma \in \Sigma} u_i(\sigma) \#_\sigma(x) = \boldsymbol{u}_i \cdot \Psi(x) \tag{23}$$

where $\#_\sigma(x)$ is the number of occurrences of $\sigma$ in $x$, and $\boldsymbol{u}_i$ denotes the vector of possible updates to counter $i$ where each index corresponds to a different $\sigma \in \Sigma$. So, $L(\boldsymbol{b}, i)$ is the linear language

$$L(\boldsymbol{b}, i) = \{x \in \Sigma^* \mid \boldsymbol{u}_i \cdot \Psi(x) - b_i = 0\}. \tag{24}$$

$\square$

Although the proof of Theorem 6.1 is nontrivial, it should be noted that $\tilde{\text{Q}}$SCL is quite a weak class. Such languages have limited ability to even detect the relative order of tokens in a string. We hope this argument might be extended to show that SCL or CL is semilinear.

## 7 CONCLUSION

We have shown that many variants of the counter machine converge to express the same class of formal languages, which supports that CL is a robustly defined class. We also proved that real-time counter languages are closed under a large number of common set operations. This provides tools for future work investigating real-time counter automata.

We also showed that counter automata are incapable of evaluating boolean expressions, even though they are capable of verifying that boolean expressions are syntactically well-formed. This result has a clear parallel in the domain of natural language, where deciding whether a sentence is grammatical is a different task than representing its deep syntactic or semantic structure. A general take-away from our results is that just because a counter machine (or LSTM) is sensitive to surface patterns in linguistic data does not mean it can build correct semantic representations. Counter memory can be exploited to weakly match patterns in language, which might provide the wrong kinds of inductive bias for achieving sophisticated natural language understanding.

Finally, we asked whether counter languages are semilinear as another way of studying their linguistic capacity. We concluded only that a quite weak subclass of the counter languages are semilinear, and encourage future work to address the general case.

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
