# OpenReview forum: "On the Linguistic Capacity of Real-time Counter Automata"
_ICLR.cc/2020/Conference — Reject_

### Official Review · AnonReviewer2 · 2019-10-22
**Official Blind Review #2**

**Rating:** 6

**Review:**

Motivated by a link between LSTMs and counter machines (suggested by recent work, e.g. Merrill, 2019 et al.), this paper studies the formal properties of counter machines (and LSTMs by extension) as grammars, in hopes of discovering why LSTMs perform particularly well in language tasks despite having no obvious hierarchical structure.

It makes the following contributions. It shows that: (1) many variants of counter machine converge to the same formal language, (2) the counter languages are closed under common set operations (e.g. intersection, union, and complement), (3) counter machines are incapable of evaluating boolean expressions, and (4) only a weak subclass of CLs are sublinear (and most are not).

While this paper gives thorough proofs, I would have liked to see more connection to practical NLP with some experiments. Also, I would have liked to see more concrete takeaways from this paper: if correctly detecting surface patterns doesn't mean that LSTMs build correct semantic representations, what can ensure that LSTMS do have a correct semantic representation?

As this paper is far from my area of expertise, I'm willing to change my score based on my co-reviewers.

**Experience Assessment:**

I do not know much about this area.

**Review Assessment: Checking Correctness Of Derivations And Theory:**

I assessed the sensibility of the derivations and theory.

**Review Assessment: Checking Correctness Of Experiments:**

N/A

**Review Assessment: Thoroughness In Paper Reading:**

I made a quick assessment of this paper.

---

> ### Author Response · Authors · 2019-11-06
> **Author Response**
>
> Thank you for your review. We agree that experiments would make the connection to modern NLP more concrete. As an initial direction, we have conducted an experiment comparing the ability of LSTMs to 1) evaluate boolean expressions and 2) verify their well-formedness. If accepted, we will include this in a revised version.
>
> Another experimental followup we have begun is evaluating pre-trained language models by assessing their ability to perform semantic evaluation of natural language. However, we believe this larger effort is beyond the scope of this paper. We think the community could benefit from our paper because it provides context for such future work, either by ourselves or others.

---

### Official Review · AnonReviewer1 · 2019-10-24
**Official Blind Review #1**

**Rating:** 1

**Review:**

The paper proof properties of counter machines that have in recent work be suggest that LSTMs can model those.

The papers starts to mentions related work that relates automaton's and counter machines with LSTMs. These related work papers do some correlational experiments partly restricted in size, layers and architecture. They provide mostly empirical evidence that some behaviour is related to performance seen in LSTMs, GRU, etc. and similar to those in counter automata.

The paper makes then the point to take  the counter machine as a simplified formal model of the LSTM. However, I would read Merrill 2019 that counter machines could be model via LSTMs but are not limited or they are not an upper bound what they can compute.  The authors does some proves on counter automata and hopes to gain insights into the properties of LSTMs used for NLP or semantic analysis and this would provide insights for the use in NLP.  It seems to me that the paper claims that counter automatons are an upper bound for the computation power of LSTMs. In the way I read this seems at least not well formulated or too strong.

I would not follow the conclusion that 'A general take-away
from our results is that just because a counter machine (or LSTM) is sensitive to surface patterns in
linguistic data does not mean it can build correct semantic representations'.
The argumentation is flawed as counter machines are not an upper limit of expressiveness of LSTMs nor do they describe well what they do. That one can use LSTMs to compute languages that counter automata can do too means not that they could do more. The property of Counter automata are useful for instance to build phrase structures meaning they can be use to express scope and keep track of. However, deeper layered networks are widely used to put structure over the scopes (arguments) to connected them in a higher order fashion.

There are many paper which show empirical how to build semantic or syntactic structures using LSTMs - also in already quite well in seq2seq fashion.

The more theoretical part looks fine to me and could be of value to readers. Nevertheless, the authors could considere to revise  their claims as they are not well supported by the evidence provided in the paper nor pervious literature cited.



**Experience Assessment:**

I have read many papers in this area.

**Review Assessment: Checking Correctness Of Derivations And Theory:**

I assessed the sensibility of the derivations and theory.

**Review Assessment: Checking Correctness Of Experiments:**

N/A

**Review Assessment: Thoroughness In Paper Reading:**

I read the paper at least twice and used my best judgement in assessing the paper.

---

> ### Author Response · Authors · 2019-11-06
> **Author Response**
>
> Thank you for taking the time to provide us with feedback.
>
> Our Theorem 5.1 shows that counter machines cannot model the semantic structures of boolean expressions, even though they can verify that they are syntactically well-formed. In our conclusion, we suggest that a similar mismatch between syntax/semantics might affect LSTMs, due to the theory in the literature that LSTM memory is structured as counter registers. We intentionally word this take-away tentatively. While tentative, we believe it is still valuable, much in line with AnonReviewer6’s comment: “If, however, this conjecture should turn out to be true, the paper would mark a very strong starting point for further exploration.” We are open to rephrasing the conclusion to emphasize its tentative nature, and that it makes predictions that future work can readily test.
>
> In light of your comments, it seems that we should clarify our discussion of why previous work motivates a connection between counter languages and the practical learnable capacity of LSTMs.
>
> As you say, there are papers showing “how to build semantic or syntactic structures using LSTMs”, but as discussed by Weiss et al. (2018) and Merrill (2019), it is not clear that gradient descent has the ability or inductive bias to reach these rigid strategies. This idea is central to the line of work that we reference in the introduction.
>
> Many works present theoretical and empirical evidence relating the behavior learnable by LSTMs to computation with counters. These include the visualization of LSTM states as counter registers (Weiss, 2018; Suzgun 2019) and empirical results showing that LSTMs fail to generalize on tasks beyond the counter languages (Hao et al., 2018; Merrill, 2019; Suzgun et al., 2019). There are also theoretical lower bound (Weiss et al., 2018) and upper bound (Merrill et al., 2019) results for asymptotic LSTMs. These results do not provably establish equivalence, but they provide a reason for thinking that studying counter automata might shed light on LSTMs.

---

### Official Review · AnonReviewer6 · 2019-11-01
**Official Blind Review #6**

**Rating:** 6

**Review:**

Summary
-------

The authors investigate (subclasses of) generalized counter machines with respect to their weak generative capacity, their ability to represent structure, and several closure properties. This is motivated by recent indications that LSTMs have comparable expressivity to counter machines, so that the formal properties of these machines might provide indirect insights into the linguistic suitability of LSTMs.


Evaluation
----------

I also reviewed this paper for SCiL a few months ago.
While I had major reservations back then, I am happy to provide a more positive evaluation this time as the authors have done some revisions that clear up many points of confusion.
I have to add two caveats, though.
First, I am a bit disheartened that the authors chose not to adopt many of the excellent changes suggested by another SCiL reviewer (who went way beyond the call of duty with their multi-page review).
Second, I did not have sufficient time to check all proofs for their correctness.
In many cases the strategies strike me as intuitively sound, but my intuition tends to miss edge cases.
Nonetheless, I think that this paper, albeit a bit of a gamble, would make for an interesting addition to the program.


1) Weakness: Link to neural networks still unclear

The central weakness of the paper is still the link between neural networks and counter automata.
Based on what is said in the paper, this is merely a conjecture at this point, not a well-established fact.
Without this link, the value of the paper is unclear.
If, however, this conjecture should turn out to be true, the paper would mark a very strong starting point for further exploration.
This makes it a gamble worth taking.


2) Strong results, but lack of examples

The results are not trivial and provide deep insights into the inner workings of counter machines.
In particular the fact that counter machines cannot correctly represent Boolean expressions reveals key limitations on their representational power.
The semilinearity result is less impressive because of how limited the machines are that it applies to, and I'm not sure that the proof provides a good basis for generalization to more complex machines.
The authors might consider removing this part to clear some space for examples, which are sorely needed.
The formalism is abstract and unfamiliar to most readers, and a few concrete examples would greatly strengthen the readers' intuition.


3) No investigation of linguistically important string languages

As the authors make claims about linguistic adequacy, it is surprising that there is no discussion of TALs, MCFLs or PMCLFs.
The grammar formalism of GPSG was abandoned because it was limited to context-free languages and could not handle those more complex language classes.
So if counter machines fail here, the issue of their linguistic adequacy is already decided without further probing semilinearity or representational power.
As far as I can tell, real-time counter machines cannot generate the PMCLF a^{2^n}, which is an abstract model of unbounded copying constructions in natural language (see Radzisnky on Chinese number names, Michaelis & Kracht on Old Georgian case stacking, and Kobele on Yoruba).
Nor is it obvious to me that counter machines can handle the copy language {ww | w \in \Sigma^*}, a model of crossing dependencies, although they can handle a^n b^n c^n (a TAL).
It should also be possible to generate the linguistically undesirable MIX language, which is a 2-MCFL but not a TAL.


Minor comments
--------------

- As noted in my SCiL review, your definitions still differ from those of Fischer et al. 1968. What is the reason for this?

- Theorem 3.1: \subsetneq would be clearer than \subset

- p4, typo: the the

- Proof of Theorem 3.2: Unless I misunderstand your modulo construction, your ICL only has resolution up to mod n. For instance, with mod 2 it can distinguish 2 from 3, but not 2 from 4. The CL can do that. Don't you need a second counter c_i' for each c_i, then, to keep track of how often you have wrapped around modulo n in c_i? That would still be incremental as you can never wrap around by more than 1 in any given update.

- Sec 6.1: in all those definitions, if should be iff


References
----------

@ARTICLE{Radzinski91,
  author = {Radzinski, Daniel},
  title = {Chinese Number Names, Tree Adjoining Languages, and Mild Context
	Sensitivity},
  year = {1991},
  journal = {Computational Linguistics},
  volume = {17},
  pages = {277--300},
  url = {http://ucrel.lancs.ac.uk/acl/J/J91/J91-3002.pdf}
}

@INPROCEEDINGS{MichaelisKracht97,
  author = {Michaelis, Jens and Kracht, Marcus},
  title = {Semilinearity as a Syntactic Invariant},
  year = {1997},
  booktitle = {Logical Aspects of Computational Linguistics},
  pages = {329--345},
  editor = {Retor{\'e}, Christian},
  volume = {1328},
  series = {Lecture Notes in Artifical Intelligence},
  publisher = {Springer},
  doi = {10.1007/BFb0052165},
  url = {http://dx.doi.org/10.1007/BFb0052165}
}

@PHDTHESIS{Kobele06,
  author = {Kobele, Gregory M.},
  title = {Generating Copies: {A}n Investigation into Structural Identity in
	Language and Grammar},
  year = {2006},
  school = {UCLA},
  url = {http://home.uchicago.edu/~gkobele/files/Kobele06GeneratingCopies.pdf}
}

**Experience Assessment:**

I have read many papers in this area.

**Review Assessment: Checking Correctness Of Derivations And Theory:**

I assessed the sensibility of the derivations and theory.

**Review Assessment: Checking Correctness Of Experiments:**

N/A

**Review Assessment: Thoroughness In Paper Reading:**

I read the paper at least twice and used my best judgement in assessing the paper.

---

> ### Author Response · Authors · 2019-11-06
> **Author Response**
>
> Thank you for acknowledging the improvements in our work, and for both rounds of reviews that you have now given us. We found the SCiL reviews extremely helpful for clarifying our arguments as well as the presentation of the proofs. If accepted, we will double-check the SCiL reviews in addition to these ones in order to incorporate feedback that we might have missed.
>
> In 1), we appreciate your point that the major question raised by our paper is a “very strong starting point for further exploration.” Along these lines, we have recently conducted some formal language experiments with LSTMs that we plan to include in a revised version. The results suggest that, empirically, learning how to evaluate boolean expressions is harder than learning to validate whether they are well-formed. We hope this makes the connection to neural networks more direct and provides a concrete starting point for future work.
>
> In 2), your suggestion to include an example illustrating the mechanics of counter machines is well taken. If accepted, we will add such a figure.
>
> For 3), you are right that a^{2^n} and {ww | w \in \Sigma^*} cannot be modeled by real-time counter automata, by similar arguments to Theorem 5.1 (boolean evaluation). We will incorporate these facts (as well as discussion of the related grammar formalisms) as a further illustration of their theoretical weakness, and possibly shorten the discussion of semilinearity. Thank you for the references.
>
> Finally, to answer your question about variation in definitions:
>
> Fischer et al. 1968 primarily focus on the non-real-time counter machines, where you can have multiple state transitions per input. Because of this, they allow a second class of states Q_a that does not consume input. If you constrain their machine to be real-time, then each state transition must consume input, so Q_a becomes unneeded, and you are left with the definition that we and Weiss et al. (2018) use. We will add something to this effect in the presentation of the definitions.

---

> > ### Comment · AnonReviewer6 · 2019-11-14
> > **Acknowledgment**
> >
> > Thanks for the detailed reply, in particular on the differences to Fischer et al 1968.

---

### Decision · Program_Chairs · 2019-12-19

**Decision:**

Reject

**Comment:**

This paper presents an analysis of the languages that can be accepted by a counter machine, motivated by recent work that suggests that counter machines might be a good formal model from which to approach the analysis of LSTM representations.

This is one of the trickiest papers in my batch. Reviewers agree that it represents an interesting and provocative direction, and I suspect that it could yield valuable discussion at the conference. However, reviewers were not convinced that the claims made (or implied) _about LSTMs_ are motivated, given the imperfect analogy between them and counter machines. The authors promise some empirical evidence that might mitigate these concerns to some extent, but the paper has not yet been updated, so I cannot take that into account.

As a very secondary point, which is only relevant because this paper is borderline, LSTMs are no longer widely used for language tasks, so discussion about the capacity of LSTMs _for language_ seems like an imperfect fit for an machine learning conference with a fairly applied bent.